# Expression of Dynorphin and Kappa-Opioid Receptors in the Bed Nucleus of the Stria Terminalis: Focus on Adolescent Development

**DOI:** 10.3390/ijms26167955

**Published:** 2025-08-18

**Authors:** Albert R. Gradev, Pavel I. Rashev, Dimitrinka Y. Atanasova, Angel D. Dandov, Nikolai E. Lazarov

**Affiliations:** 1Department of Anatomy and Histology, Medical University of Sofia, 1431 Sofia, Bulgaria; a.gradev@medfac.mu-sofia.bg (A.R.G.); adandov@medfac.mu-sofia.bg (A.D.D.); 2Institute of Biology and Immunology of Reproduction “Acad. Kiril Bratanov”, Bulgarian Academy of Sciences, 1113 Sofia, Bulgaria; prashev@ibir.bas.bg; 3Institute of Neurobiology, Bulgarian Academy of Sciences, 1113 Sofia, Bulgaria; d.atanasova@inb.bas.bg; 4Department of Anatomy, Faculty of Medicine, Trakia University, 6003 Stara Zagora, Bulgaria

**Keywords:** adolescence, BNST, drug and alcohol addiction, dynorphin, extended amygdala, kappa-opioid receptors, Wistar rats

## Abstract

The bed nucleus of the stria terminalis (BNST) is a heterogeneous and complex limbic forebrain structure, which plays an important role in drug addiction and anxiety. Dynorphin and kappa-opioid receptors (DYN/KOR) comprise a crucial neural system involved in modulating stress-induced drug and alcohol addiction. Previous studies have highlighted the BNST as a brain region with a strong DYN/KOR expression. However, no research has been conducted on the adolescent plasticity of this system. In the present study, we used 20- and 60-day-old Wistar rats to reveal the adolescent dynamics and possible sex differences of the DYN/KOR system in certain BNST nuclei associated with addiction behavior. We found a low expression of DYN in neuronal perikarya and a significant increase in DYN-containing nerve fibers in the lateral posterior and lateral dorsal nuclei of the rat BNST. In addition, an enhanced expression of KORs was observed in the examined BNST subnuclei with some sex differences favoring females, thus highlighting the importance of considering critical developmental differences between sexes in research. The dynamics of the DYN/KOR system observed in this study may help to explain the increased vulnerability of adolescents for developing drug and alcohol addiction.

## 1. Introduction

The bed nucleus of the stria terminalis (BNST) is a small and compact heterogeneous forebrain structure, which plays an important role in the negative drug withdrawal effects during abstinence [1]. It is a highly complex nucleus consisting of 11 to 18 distinct subnuclei in rodents (reviewed in [2,3]). As a key component of the limbic system, sometimes referred to as the extended amygdala, the BNST is a sexually dimorphic brain region involved in the control of sexual behavior and social dysfunction, which also comprise aggression (for recent reviews, see [4,5]). Neuroanatomical characterization studies in rodents have indicated that glutamate and GABA are the principal excitatory and inhibitory neurotransmitters in the BNST [5,6,7,8]. Growing evidence also suggests that BNST neurons contain multiple neuropeptides including opioid peptides and are innervated by many neuromodulatory systems, such as dopamine, serotonin, and histamine (reviewed by [5,9,10]).

Dynorphin (DYN), a member of the opioid peptide family, acts as a specific endogenous ligand for the kappa-opioid receptors (KOR) [11,12,13]. DYN is widely expressed throughout the anterior–posterior regions of the BNST, mainly in its oval nucleus, lateral posterior nucleus, and rhomboid nucleus [14]. However, KORs are only expressed at moderate levels in the medial posterior nucleus [15], and to a little extent in the anterolateral areas of the rat BNST [14]. These receptors have been shown to inhibit both GABAergic transmission from the central amygdala (CeA) in the oval nucleus [16] and glutamate signaling from the basolateral amygdala in the mouse BNST [17], and also reduce dopamine transmission in the nucleus accumbens [18].

The DYN/KOR system has lately been implicated in the processing of emotional and stress-related information [19], and has been considered a modulator of stress-induced addiction behavior [20]. This opioid system has been reported to be actively involved in the control of drug-seeking during abstinence and addiction development [21] and is up-regulated in the BNST during stress [22]. Recent studies in rodent models [23] and in humans with alcohol use disorder [24] have identified the BNST as a brain region responsible for anxiety-like and drug-seeking behaviors during abstinence.

Considering the key role of the DYN/KOR system in anxiogenesis and the abundance of estrogen and androgen receptors in the BNST, as well as the well-known phenomenon of a pubertal increase in drug and alcohol abuse [25], it is also important to further reveal the sex differences in the neural mechanisms mediating substance-related attitudes in adolescent physical and social development. During the adolescent period, sex hormones influence both structural brain development and sexual behavior. While the hormonal surge of puberty plays a crucial role, sexual behavior is not fully established during adolescence and certain risk inclinations such as alcohol and drug abuse may still emerge at this time [26]. Although a concept of organizational and activational effects of sex hormones in males and females exists [1,27], how these neural systems interact with sex to influence addictive behavior is still a debatable issue. Therefore, understanding the sex differences in pubertal development, expression, and function of the DYN/KOR neural system in the BNST may help explain substance-associated problems in adolescence.

In the present study, we examine the adolescent neuroplasticity of the DYN/KOR system in certain BNST subnuclei associated with behavioral responses to stressors and compare its expression in both sexes.

## 2. Results

We examined the immunohistochemical expression of DYN and the corresponding KOR in the lateral posterior (BNSTLP) and lateral dorsal (BNSTLD) nuclei of the rat BNST and, in addition, the presence of KORs in its medial posteromedial nucleus (BNSTMPM), which are actively involved in anxiety and drug and alcohol addiction.

### 2.1. Dynorphin

Immunohistochemistry revealed low-to-moderate expression levels of DYN in the BNSTLP and BNSTLD in both male and female rats of the two age groups (Figure 1). Specifically, only a few faintly stained DYN-containing neurons were observed in the BNSTLD of 20-day-old rats (Figure 1C,D), while a few intensely stained DYN-containing perikarya were scattered among the unstained neurons in this nucleus in 60-day-old males and females (Figure 1I,J). Likewise, isolated DYN-positive perikarya with a weak immunostaining were found in the BNSTLP of preadolescent animals (Figure 1E,F), while more neuronal cell bodies were seen intensely stained in the same nucleus of adult rats, predominantly in females (Figure 1K,L). Conversely, the immunohistochemical reaction for DYN was more pronounced in the nerve fibers in both BNSTLP and BNSTLD in the two sexes and age groups. In particular, numerous DYN-immunopositive varicose fibers and dot-like structures, presumably nerve terminals, were observed in both BNST subnuclei, although they were more abundant in the female adult rats (Figure 1J,L).

The quantitative analysis of DYN-immunopositive fiber density in the BNSTLD and BNSTLP subnuclei revealed clear age- and sex-dependent differences (Figure 2).

In the BNSTLD subnucleus, two-way ANOVA identified highly significant main effects of age (F(1,56) = 134.6, *p* < 0.0001) and sex (F(1,56) = 6.228, *p* = 0.0155), suggesting an independent contribution of these factors to variations in the DYN fiber density (Figure 2A). Although the interaction between age and sex was not statistically significant, post hoc analyses using Tukey’s tests confirmed a robust increase in the DYN fiber density from preadolescence (20 days) to postadolescence (60 days) within each sex. Specifically, the fiber density in 60-day-old males was significantly higher compared to 20-day-old males (mean difference = 19.50%, *p* < 0.0001). Females similarly exhibited a pronounced developmental increase in fiber density from pre to postadolescence (mean difference = 13.96%, *p* < 0.0001). Notably, postadolescent males demonstrated the highest overall density of DYN-immunopositive fibers within the BNSTLD. Thus, these results highlight a substantial developmental upregulation of dynorphinergic innervation within the BNSTLD, alongside a modest but significant modulation by sex.

A similar pattern emerged in the BNSTLP (Figure 2B). Two-way ANOVA again revealed significant main effects for both age (F(1,56) = 21.32, *p* < 0.0001) and sex (F(1,56) = 9.682, *p* = 0.0029). Post hoc tests indicated significant developmental increases in fiber density for both sexes, although with somewhat differing magnitudes. Females displayed a pronounced increase from preadolescence to postadolescence (mean difference = 6.453%, *p* < 0.01), while males exhibited a slightly smaller yet significant increase (mean difference = 4.596%, *p* < 0.05). Additionally, direct comparison between sexes at postadolescence indicated significantly higher fiber density in females compared to age-matched males (mean difference = 4.651%, *p* < 0.05). Collectively, these results confirm a clear developmental trajectory characterized by increasing dynorphinergic fiber density during adolescence, further nuanced by sex-specific differences within the BNSTLP.

Taken together, these findings underscore significant adolescent neuroplasticity within dynorphinergic circuits of specific BNST subdivisions, thus reflecting pronounced age-dependent dynamics with notable modulation by sex.

### 2.2. Kappa-Opioid Receptors

Moderate expression of KORs was seen in the BNSTLP and BNSTLD subnuclei (Figure 3). Consistent with its known role as a presynaptic receptor, KOR immunostaining was found outlining the neuronal soma. In addition, we observed a few KOR-immunoreactive neurons in the BNSTMPM of the preadolescent animals. Nonetheless, the most intense KOR immunostaining was present in the postadolescent rats, with somewhat predominant expression in the BNSTLP but not in BNSTLD of females (Figure 3). However, the BNSTMPM was the nucleus with the highest KOR expression in the BNST.

On the other hand, we found that the overall KOR expression was markedly in-creased during the adolescent period in all investigated nuclei. In fact, we registered a mild prevalence in the KOR expression in the BNSTMPM of preadolescent females (Figure 3), while after the puberty such a difference in immunostaining was not observed. In the preadolescent period, we also noted that distinct KOR immunoreactivity was present in the neuronal perikarya in the BNSTMPM, which was not found in the adults.

The quantitative densitometric analysis of KOR immunoreactivity in selected subnuclei of the BNST, i.e., the BNSTLD, BNSTLP, and BNSTMPM, revealed a significant age-related modulation of the receptor expression, measured as grayscale intensity (Figure 4). In this analysis, grayscale intensity values range from 0 (maximum immunoreactivity) to 255 (no immunoreactivity) and thus, lower grayscale values reflect higher KOR expression.

In the BNSTLD (Figure 4A), two-way ANOVA showed highly significant main effects of age (F(1,116) = 31.23, *p* < 0.0001) and sex (F(1,116) = 6.521, *p* = 0.0120), without a significant interaction between these factors (F(1,116) = 0.032, *p* = 0.859). Post hoc tests indicated a significant age-dependent increase in KOR expression, demonstrated by lower values in the grayscale intensity, for both males (** *p* < 0.01) and females (*** *p* < 0.001). In particular, mean grayscale intensity decreased from 143.31 ± 1.51 in 20-day-old males to 133.79 ± 1.62 in 60-day-old males (mean difference: 9.51 ± 2.49, *p* < 0.01), and from 139.13 ± 1.87 in 20-day-old females to 128.99 ± 2.00 in 60-day-old females (mean difference: 10.14 ± 2.49, *p* < 0.001). Although not all pairwise comparisons reached significance, the main effect of sex supports a general trend toward a higher KOR expression in females. Hence, the BNSTLD subregion exhibited robust developmental regulation of the KOR expression with an additional subtle influence of sex.

The BNSTLP subnucleus displayed a similar developmental pattern (Figure 4B). Two-way ANOVA again demonstrated a highly significant main effect of age (F(1,116) = 50.79, *p* < 0.0001), though neither sex nor the interaction between age and sex was significant. Post hoc comparisons confirmed substantial decreases in grayscale intensity, indicative of an increased KOR expression, from preadolescence to adulthood in males and females (**** *p* < 0.0001 for both). Mean grayscale values decreased from 135.35 ± 1.04 to 122.53 ± 1.37 in males (mean difference: 12.82 ± 2.32, *p* < 0.0001), and from 132.15 ± 2.08 to 121.55 ± 1.88 in females (mean difference: 10.60 ± 2.32, *p* < 0.0001). Despite a tendency toward a higher expression in females, sex was not a statistically significant factor in determining KOR levels in this region. Thus, KOR expression in the BNSTLP increased significantly and consistently with age in both sexes, with a minimal influence of sex.

In the BNSTMPM subdivision (Figure 4C), two-way ANOVA also revealed a highly significant main effect of age (F(1,36) = 42.64, *p* < 0.0001). Post hoc analyses highlighted pronounced reductions in grayscale intensity, reflecting increased KOR immunoreactivity in postadolescent males (**** *p* < 0.0001) and females (** *p* < 0.01), compared to preadolescent counterparts. Specifically, mean grayscale values decreased from 168.9 ± 3.45 to 140.5 ± 2.27 in males, and from 157.2 ± 5.36 to 137.9 ± 2.75 in females. Sex did not exert a statistically significant effect in this region, suggesting that KOR expression within the MPM subdivision is primarily regulated by developmental stage. While sex differences were not consistently significant across all comparisons, the overall pattern suggests that juvenile animals, especially 20-day-old males, exhibit the lowest receptor expression, potentially indicating a period of functional suppression of KOR signaling during early development.

Taken together, these findings demonstrate widespread and robust developmental upregulation of KOR expression across the BNST subdivisions studied, thus highlighting the significant adolescent neuroplasticity within dynorphin/KOR signaling pathways.

## 3. Discussion

A growing consensus suggests a major role for the DYN/KOR system in the BNST in maladaptive behavioral regulation related to drug and alcohol use and abuse. The present study provides immunohistochemical evidence for an intense expression of DYN in BNSTLD neurons without evident sex and age-related differences. The lack of significant interaction between these fundamental factors indicates that the differences attributed to age are consistent across sexes. It has previously been shown that DYN-expressing neurons in the BNSTLD act on the presynaptic KOR, inhibiting the afferents from the basolateral amygdala with anxiogenic properties [9,17,28]. Here, we demonstrate an increase in DYN-containing nerve fibers in the BNSTLD and also in BNSTLP during adolescence. The occurrence of immunoreactive fibers for preprodynorphin, the precursor protein for DYN, has been found in the extended amygdala, including the anterior BNST [29], BNSTLD, and BNSTLP, and it is believed that these projections most probably derive from the major output nucleus of the amygdala and the central amygdaloid nucleus, and via the stria terminalis and ansa peduncularis, they reach the BNST in adult rats [30,31]. Accordingly, it has been described that knockout of DYN in the CeA decreases alcohol consumption in both sexes in mouse models of alcohol addiction [32], whereas DYN is overexpressed in the central amygdaloid nucleus in alcohol-dependent rats [33], which makes it an important nucleus for alcohol addiction. Indeed, the significance of DYN/KOR activity within extended amygdala circuitry for stress-related alcohol intake has recently been reported in mice [34]. In this respect, CeA projections to the anterolateral BNST, which is considered part of the extended amygdala and has a significant role in addiction disorders, makes this connection important for the development of addiction disorders [35]. In addition, our immunohistochemical and semi-quantitative image analysis reveal that the density of DYN-immunostained varicose fibers is considerably increased in the BNST of adult rats when compared with preadolescent animals, although there is no statistical significance in terms of areas they occupy. These findings highlight an age-related increase in DYN-positive nerve fiber density in the two BNST subnuclei that underline a sex-dependent differentiation, especially evident in older animals. Our data also determine a robust age-related increase and moderate sex-specific variations in the density of DYN-containing fibers within the BNSTLD subnucleus, thus highlighting the importance of considering critical developmental differences between sexes in research. It is likely that an increased density of DYN-immunopositive nerve fibers may help explain the greater vulnerability of adolescents to addiction disorders with endogenous mechanisms involving the BNST [36,37].

Given the known role of the BNST in regulating sexual behavior and social dysfunction, our results further show an increased expression of KOR in certain BNST subnuclei during the adolescent period. Such an increased level of DYN/KOR expression has already been shown in the extended amygdala during alcohol abstinence [38], and, moreover, the correlation between alcohol seeking and KOR signaling in the BNST and amygdala is well proved [34,39] and recently reviewed in [19]. Indeed, previous studies have demonstrated the involvement of DYN/KOR system in pro-addictive behaviors and that the increased number of KOR receptors can enhance addiction development in humans (reviewed by [21]). Furthermore, a KOR system dysregulation in the BNST has been reported in a rat model of alcohol abstinence, thus illustrating the therapeutic potential of targeting KORs to treat alcohol dependence [40]. KORs in the BNST are also known to induce negative affective behavior in alcohol withdrawal [40]. Another proof for the KOR involvement in addiction disorders comes from the fact that the activation of KORs inhibits dopaminergic and glutamatergic transmission from different brain regions including an anxiolytic pathway from the basolateral amygdala to the BNST after stress exposure and during anxiety [17,18,41]. Although the KOR system is not widely studied specifically within the BNSTMPM in the context of drug and alcohol addiction, earlier data suggest that this membrane-bound receptor can be internalized into early endosomes as a way of receptor regulation [42,43].

It is well-known that the expression and function of DYN and KORs are sexually dimorphic, particularly in the stress and reward pathways in the brain. Prior research has shown the presence of sexual dimorphism in the expression of main transmitter systems in the rat BNST [44]. However, relatively little is known about the existence of sex differences in the DYN/KOR system in the BNST in rodents. Recent studies have revealed that the effect of KOR activation on neural transmission depends on the phenotype of neurons expressing KORs [45,46,47] and the stage of adolescent development [48]. These differences could be explained with the different exposure to sex hormones and sex differences in the expression of DYN and KORs. The adolescent increase in KOR expression has lately been suggested by Varlinskaya et al. [49], who stated that DYN/KOR system changes with age and differentially responds and adapts to stress across development. Considering our findings, we believe that the KOR adolescent increase in the expression in the BNST may predispose adolescents to develop addictions easily. Moreover, the mechanisms that have been discovered to underlie sex differences in KOR function in humans in the context of pain and mood may apply to sex differences in KOR function in other systems such as addiction and reward [47]. In this regard, sex differences in relationship between stress and drinking could somewhat explain the more frequent stress-related alcohol usage in females and could be an important future direction for developing sex-appropriate treatments in alcohol use disorder in women [50]. Taken together, these results provide support for the assumption that sex differences occur at multiple levels [51].

Finally, the adolescent increase in KOR expression can also clarify the increase in anxiety states during and after adolescence [52]. Although the effect of early postnatal stress on the DYN/KOR system is already recognized [9], adolescence stress cannot be excluded as a potential factor for KOR system changes. Future work may address these gaps in our knowledge by building on the mechanisms specifically mediating addiction in females and males.

## 4. Materials and Methods

### 4.1. Experimental Animals

The experiments were carried out on preadolescent Wistar rats (20 days-old; *n* = 12) and postadolescent Wistar rats (60 days-old; *n* = 12) of both sexes (equal number = 6 for males and females) weighing 25–40 g for preadolescent and 200–250 g for postadolescent Wistar rats, respectively. The experimental procedures were in keeping with the ethical guidelines of the EU Directive 2010/63/EU for the protection of animals used for scientific purposes, and the protocol was approved by the Bulgarian Food Safety Agency (Approval Protocol Nr. 411 of 16 December 2024). All efforts were made to minimize the number of animals used and their suffering.

### 4.2. Tissue Preparation

The animals were deeply anesthetized with an intraperitoneal injection of sodium pentobarbital (40 mg/kg; Supelco Inc., Bellefonte, PA, USA) and then transcardially perfused first with 0.05 M phosphate-buffered saline (PBS), followed by 4% paraformaldehyde in 0.1 M phosphate buffer (PB), pH 7.4. After perfusion, the brains of the animals were quickly removed, trimmed at the level of the BNST, and then postfixed in the same fixative overnight at 4 °C. Thereafter, the tissue pieces were processed for paraffin embedding. Afterwards, the paraffin blocks were cut into 6 µm thick sections which were collected sequentially and mounted on poly-L-lysine coated glass slides (Sigma-Aldrich, St. Louis, MO, USA). The first section of each series was routinely stained with hematoxylin and eosin (H&E) for histological study, and the remaining sections were used for immunohistochemistry (IHC) on the same specimen slide.

### 4.3. Immunohistochemical Procedure

Immunohistochemical detection of Dynorphin A and kappa-opioid receptor-1 (KOR-1) was performed on paraffin-embedded brain sections following standard deparaffinization procedures. For antigen retrieval, the sections were incubated in 0.01 M citrate buffer, pH 6.0, and heated in a water bath at 95 °C for 20 min (WB-4MS model, Biosan Laboratories, Inc., Warren, MI, USA). After cooling, the slides were rinsed in Tris-buffered saline with 0.05% Tween-20 (TBST) buffer (Sigma-Aldrich, St. Louis, MO, USA), pH 7.6. Endogenous peroxidase activity was blocked using 3% hydrogen peroxide in distilled water for 10 min at room temperature. To minimize non-specific background staining, sections were pretreated with Super Block solution (ScyTek Laboratories, Logan, UT, USA) for 10 min, followed by three short washes in TBST. Biotin blocking was subsequently carried out using the Biotin Blocking Kit (Cat. No. BBK120, ScyTek) in two sequential steps (Part A and Part B), each applied for 15 min with an intervening wash. Primary antibodies were diluted in Tris-based antibody diluent (ATG125, ScyTek) and applied overnight at 4 °C in a humid chamber. The following antibodies were used: rabbit polyclonal anti-Dynorphin A (1:100, Antibodies, A283838, Antibodies.com, Stockholm, Sweden) and rabbit polyclonal anti-KOR-1 (1:200, Elabscience, E-AB-64902, Elabscience, Houston, TX, USA). Thereafter, sections were incubated with the UltraTek Biotinylated Secondary Reagent and the HRP-conjugated detection reagent (UltraTek HRP Anti-Polyvalent Detection System, Cat. No. AFN600, ScyTek Laboratories, Logan, UT, USA). Visualization of the immunoreactive signal was performed using the DAB Chromogen/Substrate Kit (ScyTek Laboratories). Finally, the slides were dehydrated through a graded series of ethanol, cleared, and coverslipped.

### 4.4. Immunoreaction and Antisera Specificity Tests

We applied both positive and negative controls to test the specificity of the antibodies used in this study. For immunoreaction specificity testing, omission of the specific primary antibodies by their replacement with PBS or non-immune serum, at the same dilution as the primary antiserum, was performed and no specific immunostaining was observed under these conditions. The antibodies were further characterized with tissue from regions known to contain the antigen. Immunolabeled sections of various brain regions known to contain the antigen were used as positive controls. In addition, the specificities of the antibodies were controlled by preabsorption of the primary antisera for 2 h at room temperature or for 24 h at 4 °C with the respective synthetic antigen at a concentration of 20 or 200 μg/mL, and the preabsorbed antisera were substituted for the nonabsorbed antisera in the immunohistochemical procedure. Preabsorbed antibodies failed to stain any brain tissues.

### 4.5. Image Analysis and Statistics

After immunostaining, the specimens were examined and photographed on an Olympus research microscope. We used the standardized nomenclature system provided by atlases of the rat brain based on selections of histological sections [3,53], and the Bota and Swanson [54] table for comparison of two atlas parcellation schemes. Photomicrographs were taken from each BNST subnucleus, both from the left and right side, to increase the number of measured neurons. At least 30 neurons from the BNSTMPM were assessed. 

The digital images were obtained with an Olympus VS 120 slide scanner with a microscope acquisition system (Olympus VS-ASW 4.1.2 Image Acquisition software). Then, the color images were converted into black and white images, and we measured the intensity of grey in the grayscale. Grayscale intensity values range from 0 (black, indicating the highest receptor expression) to 255 (white, indicating no receptor expression). Thus, lower values reflect increased expression levels. These binary images represented the analyzed reaction and were further used for measurements with ImageJ 1.x software.

Quantification of DYN-positive fibers was carried out on high-resolution digital images acquired under constant microscope and camera settings. The analysis was performed using ImageJ software (NIH, Bethesda, MD, USA). Within each BNST subnucleus, a fixed-size region of interest (ROI) was delineated, and a binary threshold was applied to isolate immunopositive fibers. The density of DYN-positive fibers was calculated as the percentage of the total ROI area occupied by the DAB-labeled signal (% area).

The expression of the KOR was evaluated in specific subdivisions of the BNST, including the BNSTLD, BNSTLP, and BNSTMPM subnuclei. High-resolution digital images of immunolabeled brain sections were acquired under identical exposure conditions using a light microscope equipped with a calibrated camera system. Grayscale intensity was measured within manually delineated ROIs using standardized settings in ImageJ (NIH, USA). Pixel intensity values ranged from 0 (black, maximal signal) to 255 (white, no signal). Since signal intensity is inversely proportional to protein expression, lower grayscale values were interpreted as reflecting higher levels of KOR immunoreactivity.

Quantitative data were analyzed using Prism 9 (GraphPad Software). Normality of distribution was confirmed using the Shapiro–Wilk test. Data were subjected to two-way analysis of variance (ANOVA) with age and sex as independent factors, followed by Tukey’s multiple comparisons test for pairwise group comparisons. All results are presented as mean ± standard error of the mean (SEM). Statistical significance was accepted at *p* < 0.05.

## 5. Conclusions

In conclusion, the adolescent dynamics of the DYN/KOR system observed in several BNST subnuclei may help us explain the well-known phenomenon of increased vulnerability for developing an addiction to drugs and alcohol use in adolescents. Indeed, the increased expression of both DYN and KOR and/or the dysregulation of this system in the BNSTLD and BNSTLP which are associated with drug-seeking behavior might underlie the formation of abstinence and could be considered the morphological substrate of such an addiction hypothesis. Moreover, the reported sex differences in neural mechanisms mediating substance-related attitudes in adolescent development could further explain some notable sex differences in addiction and anxiety.

## Figures and Tables

**Figure 1 ijms-26-07955-f001:**
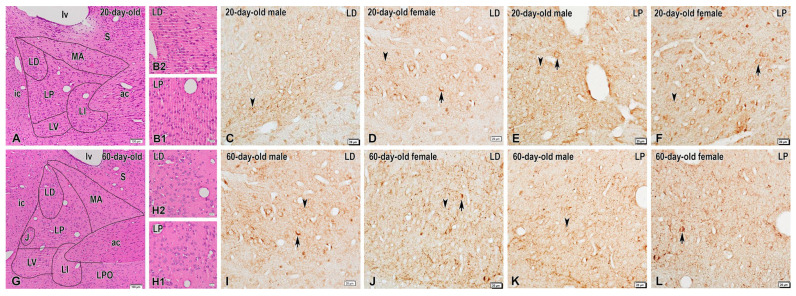
Representative images demonstrating dynorphin (DYN) expression in the bed nucleus of the stria terminalis (BNST) of preadolescent and postadolescent rats. (**A**) Low-power photomicrograph of a coronal H&E-stained section showing the structural organization of the BNST in 20-day-old rats and its composite subnuclei, i.e., medial anterior nucleus (MA), lateral dorsal nucleus (LD), lateral intermediate nucleus (LI), lateral posterior nucleus (LP), and lateral ventral nucleus (LV). Panels (**B1**,**B2**) show high-magnification images of the lateral posterior (LP) and lateral dorsal (LD) nuclei, respectively. Photomicrographs of adjacent sections illustrating DYN-immunoreactive neurons and fibers in the LD and LP nuclei of the BNST of 20-day-old male (**C**,**E**) and female (**D**,**F**) rats. Note the sparse DYN-immunoreactive neuronal perikarya (arrows) and varicose nerve fibers (arrowheads) scattered in both BNST nuclei. (**G**) Overview of BNST nuclei in 60-day-old rats, routinely stained with H&E and higher-magnification images of the LP (**H1**) and LD (**H2**) nuclei in insets. (**I**–**L**) High-power photomicrographs of the LP and LD in adult male (**I**,**K**) and female (**J**,**L**) rats showing the enhanced DYN expression in scattered neuronal somata (arrows) and varicosities (arrowheads). ac, anterior commissure; ic, internal capsule; J, juxtacapsular nucleus; LPO, lateral preoptic area; lv, lateral ventricle; S, septal area. Scale bars, 100 µm (**A**,**G**), 20 µm (**B1**,**B2**–**F**,**I**–**L**).

**Figure 2 ijms-26-07955-f002:**
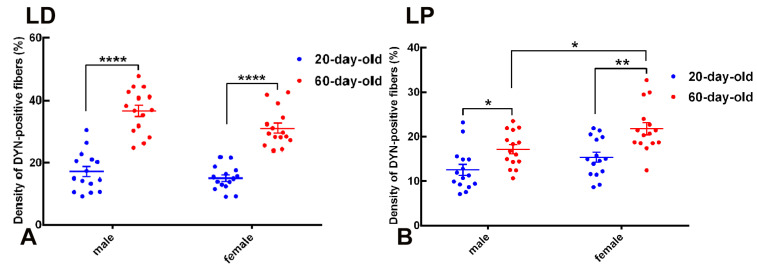
Age- and sex-dependent differences in DYN-immunoreactive fiber density in the lateral dorsal (LD) and lateral posterior (LP) subdivisions of the BNST. Scatter-dot plots illustrate quantitative analysis of the density of DYN-immunopositive fibers (expressed as the percentage of area occupied by immunoreactive fibers) in BNST subregions of 20-day-old (blue) and 60-day-old (red) male and female rats. Individual data points represent measurements from single animals, horizontal lines indicate group mean values, and vertical error bars represent the standard error of the mean (SEM). (**A**) In the BNSTLD, fiber density significantly increased from preadolescence (20 days) to postadolescence (60 days) in both sexes (**** *p* < 0.0001). (**B**) Similarly, the BNSTLP exhibited significant age-related increases in DYN fiber density, which were markedly pronounced in females (** *p* < 0.01) and statistically significant in males (* *p* < 0.05). These data reveal robust developmental upregulation and sex-specific differences in the DYNergic innervation patterns within distinct BNST subnuclei.

**Figure 3 ijms-26-07955-f003:**
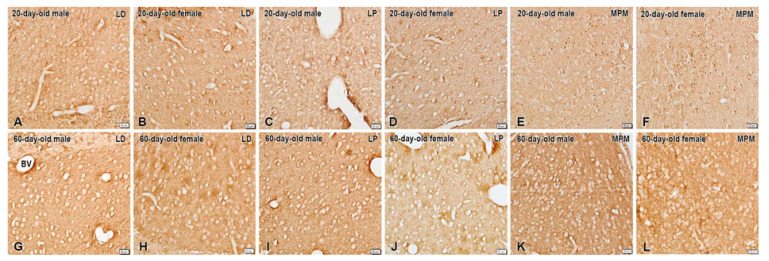
Immunohistochemical localization of KORs in the BNSTLP, BNSTLD, and BNSTMPM of male and female preadolescent and postadolescent rats. The immunostaining is more intense in the postadolescent (**G**–**L**) compared to preadolescent (**A**–**F**) subjects. The sex differences in immunostaining are only detected within the preadolescent group in the BNSTLP and BNSTMPM mainly in females, and in adults only in the BNSTLP also in females. Scale bars, 20 µm.

**Figure 4 ijms-26-07955-f004:**
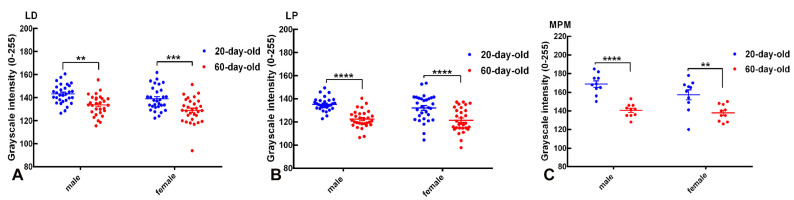
Quantitative analysis of KOR immunoreactivity in subnuclei of the BNST. Scatter dot plots illustrate grayscale intensity values (0–255), inversely proportional to the level of KOR immunoreactivity in the lateral dorsal (LD; panel (**A**)), lateral posterior (LP; panel (**B**)), and medial posteromedial (MPM; panel (**C**)) BNST subnuclei of 20-day-old (blue) and 60-day-old (red) male and female rats. Data are expressed as group means ± SEM with individual animal values indicated by dots. Lower grayscale intensity indicates higher KOR expression. Significant developmental increases in KOR immunoreactivity (decrease in grayscale intensity) were evident in the BNSTLD (males: ** *p* < 0.01; females: *** *p* < 0.001), BNSTLP (both sexes: **** *p* < 0.0001), and BNSTMPM subdivisions (males: **** *p* < 0.0001; females: ** *p* < 0.01), demonstrating pronounced age-dependent modulation across BNST subnuclei.

## Data Availability

All data generated or analyzed during this study are included in this article. Further inquiries can be directed to the corresponding author.

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
