# Peer review of "Expression of Dynorphin and Kappa-Opioid Receptors in the Bed Nucleus of the Stria Terminalis: Focus on Adolescent Development"

_ijms, 2025, doi:10.3390/ijms26167955_

Round 1
Reviewer 1 Report
Comments and Suggestions for Authors
This manuscript describes an investigation into the level of expression of the kappa opioid receptor (KOR) and its endogenous ligand, dynorphin A (DYN), in the bed nucleus of the stria terminalis (BNST). 20- and 60-day old male and female Wistar rats were used to determine the potential influence of age and sex on KOR/DYN expression. The authors found that there were significant increases in KOR/DYN expression as a function of age but not sex. This is an interesting study that raises important points about the potential role of the KOR/DYN system in adolescent addiction behaviors, and adds to the field in a positive way. There are minor edits that need to be addressed before the manuscript is suitable for publication, in my view.
The sentence in lines 67-70 ("During the tranisition...") is confusing due to the several dependent clauses connected together. Please rephrase for clarity.
The bar graphs (Figs 2 and 4) and the accompanying discussion make mention of significant differences between, for example, younger males and older females or older males and younger females. It is not clear why this is important. It seems like an odd thing to comment on, like "young men are very different from old women." There is related commentary around lines 207-210. Perhaps more convincing is needed here, or it may suffice to simply remove the significance bars and relevant discussion. A related point is that, in the conclusions section, the authors talk about a lack of significant difference between males and females of certain ages. It might be helpful to include significance bars with "n.s." for "not significant" between the groups that are discussed in the text.
Fig 3: there are several spots labeled "BV" that appear to be perhaps some sort of vesicle, but BV is not defined. Please clarify.
Author Response
We thank the reviewer for recognizing the value of our work.
Comment 1: The sentence in lines 67-70 ("During the transition...") is confusing due to the several dependent clauses connected together. Please rephrase for clarity.
Comment 2: The bar graphs (Figs 2 and 4) and the accompanying discussion make mention of significant differences between, for example, younger males and older females or older males and younger females. It is not clear why this is important. It seems like an odd thing to comment on, like "young men are very different from old women." There is related commentary around lines 207-210. Perhaps more convincing is needed here, or it may suffice to simply remove the significance bars and relevant discussion. A related point is that, in the conclusions section, the authors talk about a lack of significant difference between males and females of certain ages. It might be helpful to include significance bars with "n.s." for "not significant" between the groups that are discussed in the text.
Our response: We fully agree with the reviewer's recommendation and the proposal to present the statistical data in a more convincing way. In line with this, we have sorted out this issue by removing the unnecessary data from the statistical results. Moreover, we have rearranged this section (lines 120-148 on page 4 and lines 190-232 on pages 5 and 6), including the accompanying argument to emphasize the importance of the established in this study age-related differences in KOR expression and incorporated a brief discussion about the documented sex differences in DYN expression and the trend toward a higher KOR expression in females (lines 302-321 on page 8). Furthermore, in accordance with their recommendation we have replaced the current bar graphs (Figures 2 and 4) with scatter dot plots to better illustrate individual data and their interpretation.
Comment 3: Fig. 3: there are several spots labeled "BV" that appear to be perhaps some sort of vesicle, but BV is not defined. Please clarify.
Our response: The abbreviation “BV” stands for blood vessels which are surrounded by immunoreactive nerve fibers. However, since these are not features of the present study, we thought it appropriate to not go into detail concerning them and thus we decided to delete these labels in Figure 3.

Reviewer 2 Report
Comments and Suggestions for Authors
In the study Expression of dynorphin and kappa-opioid receptors in the bed nucleus of the stria terminalis: focus on adolescent development by Gradev et al., the authors investigate the difference in the expression of dynorphin and KOR in several subregions of BNST in early and late adolescence (20 and 60 days old) female and male Wistar rats.
While the topic of this study can be of interest to the broad readership, the MS in its current form does not meet the standards of publication in this journal. The analysis is limited and although the data presented are interesting a substantial expansion of the experimental scope and mechanistic insight would be necessary to warrant consideration for publication. Therefore, I regret to recommend rejection in its current form.
Specific comments:
In the MM section of the MS the authors incorporated the chapter 4.4 - Antisera specificity test. However, the data is missing.
Graphs should clearly display individual data points. This means ensuring that the data is presented in a way that allows viewers to easily identify and interpret each measurement.
Why were expression analyses of dynorphin in BNSTMPM not performed and analyzed?
Why was dynorphin expression measured differently than KOR expression (Density % for dynorphin and grayscale intensity for KOR)? Please provide references for both methods using DAB immunohistochemistry.
Line 256: The authors state - KOR expression in the BNSTMPM increases significantly with age, reflecting a developmental upregulation of the receptor availability that occurs in both sexes.
The presentation of data is confusing. It would be easier if on the grayscale black was assigned - 255 and white – 0, so that the presented increase and decrease of the expression on the graphs is not visually opposite from its interpretation.
Author Response
Comment 1: In the MM section of the MS, the authors incorporated chapter 4.4 - Antisera specificity test. However, the data is missing.
Our response: Thank you for pointing us in this direction. This issue is sorted out and the relevant information is instantly provided in the text, so chapter 4.4. is now renamed to Immunoreaction and Antisera specificity tests. As part of our standard validation protocol, we used both negative controls (omission of the primary antibody and substitution with PBS or non-immune serum) and positive controls to evaluate the specificity of IHC reaction and to confirm that they are not due to background noise or non-specific binding (lines 372-378 on page 9). In addition, we have performed a preabsorption test to assess how accurately the antiserum binds to the specific target antigen and the appropriate text has been added to clarify these points (lines 378-383 on page 10).
Comment 2: Graphs should clearly display individual data points. This means ensuring that the data is presented in a way that allows viewers to easily identify and interpret each measurement.
Comment 3: Why were expression analyses of dynorphin in BNSTMPM not performed and analyzed?
Our response: Dynorphin expression in the BNSTMPM was not quantitatively analyzed because previous research has indicated that dynorphin is predominantly expressed in the lateral subdivisions of the BNST (e.g., lateral posterior and lateral dorsal nuclei), with a limited or negligible presence in the BNSTMPM (cf. Poulin et al.: Progress in Neuro-Psychopharmacology and Biological Psychiatry 33(8), 2009, 1356-1365). Therefore, in our study we have focused on the expression of dynorphin in the above-mentioned subnuclei, i.e. BNSTLP and BNSTLD.
Comment 4: Why was dynorphin expression measured differently than KOR expression (Density % for dynorphin and grayscale intensity for KOR)? Please provide references for both methods using DAB immunohistochemistry.
Our response: Thank you for raising this critical methodological point. The choice of the quantification method was based on the distinct subcellular localization and immunohistochemical distribution patterns of Dynorphin (DYN) and kappa opioid receptor (KOR).
As previously described, DYN was predominantly localized within varicose axonal fibers and their terminals in the BNST (cf. Marchant et al.: J Comp Neurol 504(5), 2007, 702-715) and such a punctate, fiber-like distribution pattern is best suited for area-based quantification. Therefore, we assessed the DYN expression by measuring the percentage of immunopositively fiber area within fixed regions of interest (ROIs).
In contrast, consistent with its role as a membrane-bound signaling protein, KOR immunoreactivity was mainly observed as perisomatic or membrane-associated labeling. Such a localization is more amenable to densitometric analysis, and we thus quantified KOR expression using grayscale intensity measurements, where lower values correspond to higher receptor expression. This approach follows standard procedures for membrane receptor analysis in DAB-based immunohistochemistry, as demonstrated by Chu et al. (J Biol Chem 272(43), 1997, 27124-27130).
Hence, these complementary methods reflect current best practices for assessing immunohistochemical signal intensity depending on subcellular distribution, and this rationale is now clearly stated in the revised M&Ms section. Moreover, and as recommended, the statements in the text are now supported by appropriate references.
Comment 5: Line 256: The authors state - "KOR expression in the BNSTMPM increases significantly with age, reflecting a developmental upregulation of the receptor availability that occurs in both sexes."
Our response: According to the grayscale intensity measurements, the KOR expression was found increasing during the adolescent development in both sexes in all investigated subnuclei. In line with this, we have rearranged the result section concerning the KOR expression and added a concluding sentence (lines 230-232 on page 6) to make the issue clearer.
Comment 6: The presentation of data is confusing. It would be easier if on the grayscale black was assigned - 255 and white – 0, so that the presented increase and decrease of the expression on the graphs is not visually opposite from its interpretation.
Our response: We highly appreciate the reviewer's insightful and helpful comments and thank them for the suggestion regarding the inversion of the grayscale scale (black = 255, white = 0) for visualizing KOR expression data. However, we respectfully prefer to retain our original grayscale convention (0 = black/high signal; 255 = white/no signal), and we offer the following rationale:
Scientific and Methodological Standardization: In densitometric image analysis, it is a widely accepted convention that lower grayscale intensity values indicate higher expression levels (i.e., darker staining), whereas higher values reflect lower expression (i.e., lighter or absent staining). This standard is particularly well-established in immunohistochemistry using DAB as a chromogen and has been consistently employed (cf. Chu et al., 1997; https://doi.org/10.1074/jbc.272.43.27124; Jordan et al., 2000; https://doi.org/10.1089/104454900314672). Maintaining this convention ensures methodological consistency, facilitates interstudy comparisons, and enhances reproducibility across related research.
Technical Justification and Consistency: Digital images obtained via microscopy-based slide scanners typically encode pixel intensity following the standard grayscale format, where 0 represents maximum intensity (black) and 255 represents minimum intensity (white). Using the standard encoding convention avoids additional, unnecessary data transformation steps, thereby maintaining data integrity and clarity.
Standard Practice in Scientific Literature: Recent studies in the fields of neuroanatomy and neuroscience demonstrate adherence to the grayscale intensity convention we employed, where lower grayscale values correspond to higher signal intensity (cf. Rose et al., 2016; https://doi.org/10.1093/ijnp/pyv127; Crowley et al., 2016; https://doi.org/10.1016/j.celrep.2016.02.069). Departing from this established convention solely for our manuscript could lead to inconsistency with existing literature and may confuse researchers accustomed to the standard approach.
Taking into account methodological standardization, data integrity, alignment with established literature, and the importance of clarity, we respectfully propose to retain our original grayscale approach. To ensure an intuitive interpretation of the data, we have supplemented the revised manuscript with clear explanatory notes describing the grayscale scale and its interpretation.
Thank you again for your thoughtful comments and suggestions that help us clarify and strengthen our manuscript. If there are any additional points to be clarified, please do not hesitate to let us know in due course.

Round 2
Reviewer 2 Report
Comments and Suggestions for Authors
The authors have successfully responded to all my comments.